Computational analysis of the LRRK2 interactome

Manzoni Claudia 1 2 c.manzoni@reading.ac.uk
Denny Paul 3
Lovering Ruth C. 3
Lewis Patrick A. 1 2 4
1 School of Pharmacy, University of Reading , Whiteknights, Reading , UK
2 Department of Molecular Neuroscience, UCL Institute of Neurology, University College London , Queen Square, London , UK
3 Institute of Cardiovascular Science, University College London , London , UK
4 Centre for Integrated Neuroscience and Neurodynamics, University of Reading , Whiteknights, Reading , UK
Perez-Acle Tomas
Electronic publication date: 2015 Feb 19
Publication date: 2015
Volume: 3
Electronic Location ID: e778
Received 2014 Nov 5; Accepted 2015 Jan 29
Copyright: © 2015 Manzoni et al.
Copyright year: 2015
Copyright holder: Manzoni et al.
License: This is an open access article distributed under the terms of the Creative Commons Attribution License, which permits unrestricted use, distribution, reproduction and adaptation in any medium and for any purpose provided that it is properly attributed. For attribution, the original author(s), title, publication source (PeerJ) and either DOI or URL of the article must be cited.
License URL: https://creativecommons.org/licenses/by/4.0/

Keywords: LRRK2, Interactome, Protein–protein interactions, Parkinson’s disease, GO terms enrichment

Funding: Michael J. Fox Foundation Parkinson’s UK F1002 Rosetrees Trust Wellcome Trust/MRC Joint Call in Neurodegeneration WT089698 MRC New Investigator Research Grant MR/L010933/1 Parkinson’s UK G1307 National Institute for Health Research University College London Hospitals Biomedical Research Centre Research support was provided by the Michael J. Fox Foundation, Parkinson’s UK and the Rosetrees Trust. Patrick A. Lewis is a Parkinson’s UK research fellow (grant F1002). This work was supported in part by the Wellcome Trust/MRC Joint Call in Neurodegeneration award (WT089698) to the UK Parkinson’s Disease Consortium (UKPDC) whose members are from the UCL Institute of Neurology, the University of Sheffield and the MRC Protein Phosphorylation Unit at the University of Dundee, and by a MRC New Investigator Research Grant (MR/L010933/1) to Patrick A. Lewis. Paul Denny and Ruth C. Lovering are supported by a Parkinson’s UK grant (G1307) and by the National Institute for Health Research University College London Hospitals Biomedical Research Centre. The funders had no role in study design, data collection and analysis, decision to publish, or preparation of the manuscript.

==============================
LRRK2 was identified in 2004 as the causative protein product of the Parkinson’s disease locus designated PARK8. In the decade since then, genetic studies have revealed at least 6 dominant mutations in LRRK2 linked to Parkinson’s disease, alongside one associated with cancer. It is now well established that coding changes in LRRK2 are one of the most common causes of Parkinson’s. Genome-wide association studies (GWAs) have, more recently, reported single nucleotide polymorphisms (SNPs) around the LRRK2 locus to be associated with risk of developing sporadic Parkinson’s disease and inflammatory bowel disorder. The functional research that has followed these genetic breakthroughs has generated an extensive literature regarding LRRK2 pathophysiology; however, there is still no consensus as to the biological function of LRRK2. To provide insight into the aspects of cell biology that are consistently related to LRRK2 activity, we analysed the plethora of candidate LRRK2 interactors available through the BioGRID and IntAct data repositories. We then performed GO terms enrichment for the LRRK2 interactome. We found that, in two different enrichment portals, the LRRK2 interactome was associated with terms referring to transport, cellular organization, vesicles and the cytoskeleton. We also verified that 21 of the LRRK2 interactors are genetically linked to risk for Parkinson’s disease or inflammatory bowel disorder. The implications of these findings are discussed, with particular regard to potential novel areas of investigation.

Introduction

LRRK2 is the most frequently mutated gene in familial Parkinson’s disease (PD) (Paisán-Ruíz et al., 2004; Zimprich et al., 2004), and has also been identified as a risk locus for the sporadic form of the disease (Nalls et al., 2014). There are additional reports implicating allelic variants at the LRRK2 locus with increased risk of developing inflammatory bowel disorder (IBD) (Jostins et al., 2012). LRRK2 has also been linked to susceptibility to multibacillary leprosy (Zhang et al., 2009), although reproducibility of the data is controversial (Marcinek et al., 2013; Zhang et al., 2011). Furthermore, genetic as well as epidemiological data supports a role for LRRK2 in cancer for certain populations (Ruiz-Martínez et al., 2014; Inzelberg et al., 2012; Saunders-Pullman et al., 2010). The protein product of the LRRK2 gene, Leucine Rich Repeat Kinase 2 (LRRK2) is a large enzyme hosting kinase and GTPase functions surrounded by protein–protein interactions domains (Greggio et al., 2008). It is now well recognized that there are complex regulatory events linking the two enzymatic activities within LRRK2. These include, although are likely not to be limited to, regulation of dimerization by GTP and/or GDP binding and auto-phosphorylation events (Taymans et al., 2011). The importance of LRRK2 in a number of human diseases has motivated many different groups to try and solve the conundrum of its pathophysiological activity. This has led to a large and increasing body of information emerging from functional studies of LRRK2. Despite the bulk of existing data, there is no consensus on the phosphorylation targets of the LRRK2 kinase domain, or with regard to the binding partners with which LRRK2 interacts. Many different functions have been reported to be associated with LRRK2, ranging from cytoskeleton organization to vesicle trafficking, from synaptic activities to autophagy, from mitochondria homeostasis to protein synthesis, involving multiple different signalling cascades such as the m-TOR and Wnt pathways (Paisán-Ruiz, Lewis & Singleton, 2013). None of these have been categorically proven; the only reproducible data at the moment appears to be the interaction between LRRK2 and 14-3-3 that has been shown to regulate LRRK2 localization in the cell (Dzamko et al., 2010). The reasons for this volume of different and sometimes contradictory literature may be the result of variable reproducibility across the functional models available to study LRRK2; the not yet proven and sometimes axiomatic assumption that the physiological function of LRRK2 can be inferred via its pathology and vice versa; and finally the difficulties encountered studying an enzyme possessing two potentially independent activities that may link to independent cellular functions (Lewis & Manzoni, 2012).

In the past decade, a number of databases have been established to collect protein annotations from the published literature and high-throughput datasets (Hermjakob et al., 2004; Breitkreutz, Stark & Tyers, 2003; Ashburner et al., 2000). These repositories store gene and protein data in easily accessible, standardized formats, allowing comparisons between datasets, ontological groups, genes, and gene/protein families. In addition, they can be used to infer new data by deriving them from statistical analysis of gene/protein sets (Orchard , 2012).

For a protein such as LRRK2, where the volume of functional literature exceeds the capacity of researchers to analyse on an ad hoc basis (>1200 PubMed entries as of 30th October 2014) and where the range of techniques and models can cloud interpretation, a bioinformatics approach using freely available, manually-curated protein–protein interactions (PPI) datasets provides a powerful tool to develop a picture of LRRK2 in the cell environment and infer functional implications.

Herein we describe an in silico investigation of the LRRK2 interactome. Taking advantage of the manually curated datasets stored in the BioGRID and IntAct repositories (Hermjakob et al., 2004; Breitkreutz, Stark & Tyers, 2003), we prepared a filtered list of potential LRRK2 interactors, sorted them into heterologous interactions (LRRK2-human protein), which were used for GO terms enrichment, and into homologous interactions (LRRK2–LRRK2), analysed to shed light on LRRK2 dimerization.

Materials and Methods

LRRK2 interactors: IntAct database

http://www.ebi.ac.uk/intact/

The IntAct database (Kerrien et al., 2007; Hermjakob et al., 2004) was queried for LRRK2 (human, Q5S007). The list of interactors was filtered to remove the spoke expanded co-complex (i.e., annotations derived by an expansion algorithm sampling co-complexes data into binary interactions and therefore listing proteins which may not interact directly with LRRK2). The MI-TAB 2.7 file was downloaded (26th July 2014) and imported into an Excel spreadsheet. Filter #1 screened the taxid to retain only human–human interactions (annotation removed for non-human taxid or chemicals). Filter #2 was applied to remove multiple entries (in the same publication) when: (i) Interactor A was used as bait in the first experiment and as prey in the second one and interactor B vice versa; (ii) the same experiment was performed using differently tagged A and B interactors (e.g., GST Vs FLAG Vs HA). To apply filter #2, the following cut-off was established: 2 entries with the same “Publication Identifier(s)” and “#ID(s) interactor A and ID(s) interactor B”, were considered duplicated (i.e., one of them to be removed) if “Interaction detection method(s)” AND “Host organism(s)” were identical.

As a general note, protein fragments can be retrieved with the same UniProtKB identifier as for the entire protein (e.g., MAP1B catalytic domain (P46821-PRO_0000018605) has the same identifier as the entire MAP1B protein (P46821)); protein isoforms are queried by the same UniProtKB identifier as for the principal isoform (e.g., DVL2 isoform 2 (O14641-2) has the same identifier as DVL2 (O14641)). For this reason no discrimination was made between protein fragments and the entire protein, nor between different protein isoforms and the principal isoform.

LRRK2 interactors: BioGrid database

http://thebiogrid.org/

BioGrid database (Chatr-Aryamontri et al., 2013; Breitkreutz, Stark & Tyers, 2003) was queried for LRRK2 (human, Q5S007). The list of interactions was downloaded (12th August 2014) as a TAB 2.0 file and imported into an Excel spreadsheet to be analysed. Filters #1 and #2 were applied, as described above for the processing of the IntAct dataset. Filter #1 removed non-human interactors, and filter #2 removed multiple entries with the same experimental setting.

Enrichment analysis

GO enrichment analysis has been performed by the use of 2 different web-based applications.

WebGestalt (analysis on 20th August 2014, WebGestalt was last updated on 30th January 2013). (Wang et al., 2013; Zhang, Kirov & Snoddy, 2005) http://bioinfo.vanderbilt.edu/webgestalt/ was used with the following parameters: Reference set for enrichment analysis: h.sapiens_genome; Enrichment: GO analysis; Datasets: biological process and cellular component. Analysis was performed using hypergeometric statistics and Bonferroni’s correction for multiple testing. The minimum number of genes was set at 2 and the significance level was set to retrieve the top10 hits. All the input proteins were mapped. GO enrichment (supported by Panther 9.0, released 20 January 2014; analysis on 19th August 2014) (Mi, Muruganujan & Thomas, 2012) http://www.geneontology.org/page/go-enrichment-analysis was run with the following parameters: Species: human, Ontology: biological process. The statistical analysis included Bonferroni’s correction for multiple testing. All the input proteins were mapped. Since the enrichment in WebGestalt was for p-values between 10−9 and 10−7, an arbitrary p-value threshold was set at 10−7 as cut-off for the Panther enrichment.

Comparison between LRRK2 interactome and GWAs hits

PD GWAs meta-analysis (Nalls et al., 2014) was use to extract SNPs associated with sporadic PD (22 SNPs were from discovery phase and replication, 2 SNPs were from replication of previous results (Do et al., 2011; IPDGC &WTCCC2, 2011). A total of 163 IBD risk associated SNPs were extracted from Jostins et al. (2012). The position of each positive SNP was retrieved in Entrez SNP using the GRCh38 genome build; genes within an interval of ±200 Kbp were listed. The entire set of genes was then queried against the LRRK2 interactome. Positive matches were extracted and the distance between the associated SNP and the coding region was estimated.

Software

Data were stored and analyzed as Excel spreadsheet files. Graphs were prepared by the use of Prism (GraphPad) and Cytoscape (2.6.2, freely available online at http://www.cytoscape.org/) (Saito et al., 2012; Shannon et al., 2003).

Results

LRRK2 interactors: IntAct and BioGRID merged dataset

The IntAct database was queried for LRRK2 (human, Q5S007), and 542 unfiltered annotations were downloaded. After applying filter #1 to include only human interactions and filter #2 to remove multiple annotations with similar experimental details in the same publication (see materials and methods), the total number of annotations dropped from 542 to 307. Of these 307 annotations, 278 described interactions between LRRK2 and different partners (heterologous interactions) and 29 annotations described LRRK2 self-interaction (homologous interaction).

The BioGRID database was similarly queried, and 260 unfiltered annotations were downloaded. After applying filter #1 and filter #2 the total number of annotations dropped from 260 to 230, of which 194 were heterologous and 36 homologous interactions.

Detailed analysis of the PubMed IDs for the heterologous interactions in the IntAct and BioGRID filtered datasets revealed that annotations came from 43 publications in IntAct and 42 publications in BioGRID. However, only 22 publications were identical in both datasets. Thus, considering a total of 63 annotated papers, 35% of them were annotated by both IntAct and BioGRID; 65% were contained either in IntAct (21 publications) or BioGRID (20 publications). For this reason the IntAct and the BioGRID datasets for LRRK2 heterologous interactions, after the previously described filtering, were merged to prepare the final dataset for heterologous interactions only (hereafter referred to as “merged dataset”). In the case of publications duplicated between IntAct and BioGRID only one set of annotations was moved to the merge dataset. Discrepancies were found for some of the shared records; in particular, the number of annotations was not necessarily consistent between IntAct and BioGRID, this has been previously reported (Lehne & Schlitt, 2009) and is likely due to differences in the technical classification of the experiments. To overcome this problem, in the case of shared publications, the record containing the major number of annotations was selected to be moved into the merged dataset; however, for 6 shared publications differences were as such that a merge of the two records was necessary (details of these 6 problematic records and how they were merged are in Table S1). The final, merged dataset for human LRRK2-human protein interactions (heterologous interactions) consisted of 63 publications describing 422 annotations, captured via different PPI detection methods (Fig. 1), for a total of 269 LRRK2 binding partners (here after referred to as the “complete” LRRK2 interactome).

Figure 1 LRRK2 heterologous interactions as reported in the merged dataset.

LRRK2 in the middle of the graph is linked to candidate partners (black dots) through 422 annotations as described in the merged dataset. Some partners have one connection only; others have multiple connections based on the number of annotations. Different colours represent different methods used to infer the interaction; note that since IntAct and BioGRID use different classifications for the interaction detection method, a simplified and harmonized version has been applied to this figure to help the reader. In particular: (i) Affinity Capture stands for—Affinity Capture-Western and Affinity Capture-MS (in BioGRID)—Anti Bait Coimmunoprecipitation, Anti Tag Coimmunoprecipitation, Coimmunoprecipitation, Pull Down and Affinity Chromatography Technology (in IntAct); (ii) Biochemical Activity stands for—Biochemical Activity (in BioGRID)—Protein Kinase Assay (in IntAct); (iii) Co-localization stands for—Co-localization (in BioGRID)—Confocal microscopy and Fluorescence Microscopy (in IntAct); (iv) Reconstituted Complex stands for—Fluorescence Polarization Spectroscopy and Surface Plasmon Resonance (in IntAct).

The 422 annotations in the merged dataset were scored based on the following:

Low occurrence—the protein identifier was reported just once in the list of 422 annotations.

Medium-low occurrence—the protein identifier was reported in 1 publication but with 2 or more experimental approaches.

Medium-high occurrence—the protein identifier was reported in 2 or more publications but with the same experimental technique.

High occurrence—the protein identifier was reported in 2 or more publications with 2 or more experimental approaches.

This ranking system allowed us to classify the interactors as follows: 207 interactors were annotated only once; 24 LRRK2 interactors were reported in 1 publication only but with different methods; 13 LRRK2 interactors were reported in more than 1 publication but with the same detection method; finally, 25 LRRK2 interactors were reported in more than 1 publication and with different experimental techniques. Interactors annotated only once are detailed in Table S2. We designated interactors that were reported with 2 or more annotations to represent the “LRRK2 interactome,” this dataset contains 62 proteins (Fig. 2) and is the dataset that was used for enrichment analysis.

Figure 2 Number of annotations capturing each of the LRRK2 interactors.

Only the interactors reported in 2 or more annotations, and used in the enrichment analysis, are included in the figure, all other interactors were identified in just a single experiment. (A) High occurrence LRRK2 interactors, >1 publication, >1 method; (B) Medium-high occurrence LRRK2 interactors, >1 publication, 1 method; and (C) Medium-low occurrence LRRK2 interactors, 1 publication, >1 method.

The 62 proteins in the LRRK2 interactome were segregated into protein family groups according to their UniProtKB record “family&domains” (http://www.uniprot.org/uniprot/). Twenty-two of the proteins could be assigned to 10 different families: heat shock protein 90 family; tubulin (alpha and beta) family; argonaute family; small GTPase superfamily, Rho family; STE Ser/Thr protein kinase family; 14-3-3 family; TRAFAC class dynamin-like GTPase superfamily, dynamin/Fzo/YdjA family; DSH family, small GTPase superfamily, Rab family and actin family (Fig. 3).

Figure 3 Families of LRRK2 interactors.

Each of the 62 proteins in the LRRK2 filtered interactome was associated to a family (if available) based on the UniProtKB “family&domains” classification. Proteins belonging to the same family were plotted accordingly to the total number of publications (y-axis); each family was then classified based on the authorships as non-independent (different family members described by the same research group) or independent (different family members described by at least two different research groups).

GO: “biological process” and “cellular component” enrichment analysis (WebGestalt)

The online platform WebGestalt was used to conduct a GO term enrichment analysis for the 62 LRRK2 interactors described in Fig. 2. The enrichment settings allowed for retrieval of the first top10 hits. The enriched “GO biological process” terms identified were related to transport/localization and cell organization, with a couple of terms supporting an involvement of the LRRK2 interactome in regulating kinase activity (Fig. 4). For the enriched “GO cell component” terms, as expected, the majority of the LRRK2 interactors were annotated as located in the cytoplasm or cytosol-associated components. However, on a more specific level, the interactors were clustered to vesicles, cytoskeleton and cell projections (Fig. 5). The enriched terms were sorted in GO term groups. Groups were identified as cell organization, transport/localization, regulation of kinase activity, cytosol, cytoskeleton, vesicles and cell projections (Table S3). The WebGestalt enrichment portal returns the break-down of proteins contributing toward the enrichment of each single term (Data S1). Therefore LRRK2 interactors were listed as associated to the groups for which they contributed toward enrichment. The lists were then compared to calculate the percentage of proteins in the LRRK2 interactome to contribute to each single GO term group and the percentage of intersection between groups (Fig. 6). Nearly half of the LRRK2 interactors fall in the groups of transport/localization, cytoskeleton and cell organization. Nearly 30% of them were associated with vesicles and cell projections. As expected, each of the GO term groups presented with an intersection of around 100% with the GO term cytosol group. Of more interest is that 50% of interactors associated with the regulation of kinase activity were shared with vesicles, and 72% with transport/localization thus suggesting a possible role of LRRK2 in controlling kinase activities related to vesicle/membranes and transport processes. In addition, 79% of interactors associated with vesicles were also associated with transport/localization, thus implying a possible role of LRRK2 in vesicle dynamics related to intracellular transport/trafficking.

Figure 4 Dendrogram of “GO biological process” terms enriched for LRRK2 interactors in WebGestalt.

Hierarchical levels of the dendrogram are alternatively represented in blue and yellow; red text indicates the top 10 GO terms. The table lists details of the enriched terms: GO term and ID, number of proteins in the GO term category (Ref.), number of LRRK2 interactors associated with the GO term (Genes), p-value adjusted for multiple testing.

Figure 5 Dendrogram of “GO cell component” terms enriched for LRRK2 interactors in WebGestalt.

Hierarchical levels of the dendrogram are alternatively represented in blue and yellow; red text indicates the top 10 GO terms. The table lists details of the enriched terms: GO term and ID, number of proteins in the GO term category (Ref.), number of LRRK2 interactors associated with the GO term (Genes), p-value adjusted for multiple testing.

Figure 6 LRRK2 interactors enrichment for “GO cell component” and “GO biological process” terms.

The LRRK2 filtered interactome is shown after WebGestalt enrichment analysis. The enriched GO terms were grouped in: transport/localization, cell organization, regulation of kinase activity, cytosol, vesicles, cytoskeleton and cell projections. Every protein in the filtered interactome was connected to the GO term groups it has participated toward the enrichment. In the table the 7 groups are listed in columns (A–G) and rows (1–7). The percentage of proteins from the LRRK2 interactome that contributed toward the enrichment of the GO term group listed in the row and that were also reported in the GO term group reported in the column was calculated (intersection between the group in the row and the group in the column). In the last row of the table the percentage of the proteins that contributed toward the enrichment of each GO term group was calculated against the total number of LRRK2 interactors (i.e., 62 proteins). Cells discussed in the text are highlighted.

GO: “biological process” enrichment analysis (Panther)

The filtered list of 62 LRRK2 interactors, was also used for a second, independent enrichment analysis using the GO Consortium term enrichment service, supported by Panther (Mi, Muruganujan & Thomas, 2012). The analysis retrieved 150 “GO biological process” enriched terms. These terms were manually divided into 13 groups: general terms, transport/localization, membrane processes, signalling, regulation of catalysis, metabolism, catabolism, regulation of kinase activity, regulation of mitochondrion organization, cell death, development, cell organization and immune response (complete list of terms, divided in groups, with p-values and sample frequencies is reported in Table S4). The terms classified within the general terms group were excluded from following analyses because of their lack of specificity; the terms present in the 12 remaining GO term groups were ranked based on their p-values in significance levels from 10−16 to 10−7. The contribution of each functional group toward the enrichment in a particular significant level was calculated as follows (function 1): % functional group G in significance level N=∑i=1gi*100∑j=1nj.

Function 1

Where g is the number of terms enriched in the functional group G within the significance level N; n is the total number of enriched terms across all the functional groups within the significance level N. Results are shown in Fig. 7. The significant levels 10−16, 10−15, 10−14 and 10−12 were only composed of terms belonging to the groups of transport/localization, cell organization and membrane processes. Once these highly significant terms were taken aside, all the other GO term groups started to present their contributions toward the enrichment (in the significant levels from 10−11 to 10−7).

Figure 7 “GO biological process” terms enriched for LRRK2 interactors in GO supported by Panther.

Pie charts showing the composition of every significance level of enrichment. The legend shows the 12 GO term groups, although in the charts the group name was reported only for those that reached the 10% contribution toward the enrichment.

LRRK2 interactome: IntAct and BioGRID merged dataset for LRRK2 self-interactions

The IntAct dataset contained 29 annotations for LRRK2 self-interactions (homologous interactions); BioGRID, a total of 36 homologous interactions. Only 7 publications were shared between the two datasets therefore, in the total of 31 publications reporting LRRK2 self-interaction, 77% of the informfation was captured independently by IntAct or BioGRID. Following the same procedure used for the heterologous interactions, a merged dataset was prepared, with a final count of 31 publications and 53 annotations for LRRK2 self-interaction, based on different experimental methodologies (Fig. 8).

Figure 8 Annotations for LRRK2 self-interaction.

LRRK2 in the middle of the graph is linked to the first author of publications describing self-interaction. Different colours represent different methods used to infer the interaction; note that since IntAct and BioGRID use different classifications for the interaction detection method, a simplified and harmonized version has been applied to this figure to help the reader. In particular: (i) Affinity Capture stands for—Affinity Capture-Western (in BioGRID)—Anti Tag Coimmunoprecipitation, and Pull Down (in IntAct); (ii) Biochemical Activity stands for—Biochemical Activity (in BioGRID)—Protein Kinase Assay (in IntAct).

The IntAct database (not BioGRID) reports details (when available) on the regions of the LRRK2 protein participating in protein interactions. Based on this information, a profile of the LRRK2 fragments/residues associated with LRRK2 self-interaction was prepared (Fig. 9). As expected, the catalytic core of LRRK2 was most frequently reported to be associated with LRRK2 dimerization (with a top score of 17 publications), and all the residues tested directly for self-interaction lay in the catalytic core of LRRK2.

Figure 9 Profile of LRRK2 self-interaction.

Each residue described as involved in LRRK2 self-interaction is reported with a dot, the dimension of the dot is proportional to the number of publications annotating that specific residue. A profile covering the entire length of the LRRK2 protein is shown in blue, in the bottom half of the image; the y value of the profile represents the number of publications in which the fragment in the profile was reported as associated with LRRK2 self-interaction.

LRRK2 interactome and disease

The complete LRRK2 interactome comprised of 269 proteins (the filtered interactome combined with the interactors identified by 1 annotation only) was compared with data from PD and IBD GWAs. Since the associated SNP is only a marker for a genomic locus, not an indication of a specific gene, all the coding open reading frames (ORF), for a total of 156 ORFs in the interval ±200 Kbp from the PD associated SNPs, have been taken into account. The positive matches between GWAs hits and LRRK2 interactors were recorded (Table 1). Four proteins in the LRRK2 interactome (SNCA, RAB7L1, GAK and MAPT) were found to be candidate proteins in the PD GWA. These proteins were annotated multiple times as LRRK2 interactors from different publications. Among these proteins, SNCA is not only associated with the risk of sporadic PD, its mutations are also responsible for inheritance of familial PD. A total of 1,004 ORFs were identified within a 200 Kbp region on either side of the associated SNPs for IBD and 17 proteins in the LRRK2 interactome were found to be positive matches, 4 out of 17 were indicated as candidate proteins in the GWA study, the others were just around the associated SNP; the LRRK2 interactors matching the IBD GWA, with the exception CDC37, were annotated in just 1 publication.

Table 1 LRRK2 interactors associated to the genetics of PD or IBD.

The approximate distance in base pair between the associated SNP and the protein coding sequence was calculated; the protein is indicated as candidate protein according to the original GWAs study.

Protein	A	P	Methods	HTP	Familial
PD	GWA
PD	GWA
IBD	Associated
SNP	Distance
(Kbp)	Candidate	
SNCA	6	2	IP, CM	−	X	X		rs356182	∼20	Yes	
RAB7L1	5	1	PA, IP, FM, CM	+		X		rs823118	∼14	Yes	
GAK	5	2	PA, IP, FM, CM	+		X		rs34311866	∼25	Yes	
MAPT	7	2	IP, BA	−		X		rs17649553	0	Yes	
CDC37	5	4	MS, IP	+			X	rs11879191	0	No	
DVL1	4	1	IP, CM, 2H	+			X	rs12103	∼23	No	
GNA12	1	1	MS	+			X	rs798502	0	Yes	
CALM3	1	1	MS	+			X	rs1126510	∼9.7	Yes	
GLTPD1	1	1	PA	+			X	rs12103	∼13	No	
CEP72	1	1	PA	+			X	rs11739663	∼18	No	
ARPC2	1	1	MS	+			X	rs2382817	∼32	Yes	
RIPK2	1	1	BA	−			X	rs7015630	∼73	Yes	
PSMG1	1	1	PA	+			X	rs2836878	∼82	No	
DBN1	1	1	MS	+			X	rs12654812	∼89	No	
RPAP3	1	1	PA	+			X	rs11168249	∼108	No	
CD2BP2	1	1	PA	+			X	rs11150589	∼116	No	
PLK1	1	1	PA	+			X	rs7404095	∼163	No	
ACTR2	1	1	MS	+			X	rs6740462	∼169	No	
STIP1	1	1	PA	+			X	rs559928	∼178	No	
MYO1B	1	1	MS	+			X	rs1517352	∼179	No	
CLTC	1	1	MS	+			X	rs1292053	∼189	No	
Notes.

A number of annotations

P number of publications

HTP one of the detection methods is high throughput

Methods are as followsIP immunoprecipitation

CM confocal microscopy

FM fluorescent microscopy

PA protein array

BA biochemical activity

MS affinity capture mass spectrometry

2H two hybrid

Discussion

The implication of LRRK2 in different human diseases has made this protein the centre of interest for many research groups, leading to a large number of functional studies. The breadth of the LRRK2 literature is complemented by its depth, with a wide range of different approaches and techniques used in investigations. A PubMed search for LRRK2 retrieved 1257 publications on the 30th October 2014. As a consequence of this, LRRK2 is an excellent candidate for in silico analysis to critically appraise the literature creating a synthesis of our understanding of the LRRK2 interactome. LRRK2 literature can be divided into two different blocks, the first one comprises functional publications that study LRRK2 without reporting PPI data, the second contains a smaller number of publications with details about physical interactions between LRRK2 and partners. This second set of publications is what is annotated in PPI databases. The critical collection of the LRRK2 interacting proteins can be used to generate the state of the art LRRK2 interactome. The interactome can be then analysed to rationalize the understanding of the existing functional literature, which describe the role of LRRK2 in a multitude of biological processes (not annotated in PPI databases). The result of such analysis can also suggest possible future wet-lab investigations.

When searching for LRRK2 interactions, we found that the information stored in the IntAct and BioGRID databases overlapped only for around 50%; this is probably a consequence of the curator’s choice in paper selection and the database specialization. Moreover IntAct and BioGRID are part of the IMEX consortium and thus committed to minimize redundant annotations (Orchard et al., 2012). Therefore, to have a more detailed view of the LRRK2 interactome the two datasets were merged. After filtering out interactions with non-human partners and repeated annotations, 422 entries, corresponding to 269 different interactors, were found to describe the LRRK2 interactome. Only 23% of the interactors (62 interactors) were confirmed with 2 or more annotations coming from either different publications or different experimental approaches; the percentage was reduced to 9.3% (25 interactors) when the selection criteria were strengthened allowing confirmation in at least 2 different publications and by using at least 2 different techniques. It is noteworthy that LRRK2 is annotated frequently as self-interaction partner. This has been reported in 31 publications and classified with 54 entries. After analysis of the LRRK2 interactors that have been confirmed with at least 2 annotations, 53% of the interactors fall into 10 protein families; 8 of those were identified independently by different research groups. This first result has at least 2 consequences. First, it suggests that, based on the features of the annotations stored in databases, not all entries are equal. Future functional research should therefore be oriented toward the interactors with stronger experimental evidence, while more biochemical investigation is needed to test the others, to confirm or discard them. Secondly, by looking at the methods used to retrieve the interactions, it has to be taken into account that many annotations come from hypothesis driven approaches: if a protein was previously mentioned, it is more likely that other researchers will plan to investigate it. This results in a degree of ascertainment bias: when a protein is identified as probable interactor, then it is more frequently examined and therefore annotated. This may influence the gathering of annotations around a specific family. Frequently annotated proteins are probably reliable interaction partners. Many interactors are annotated just once; we have considered them less reliable since they have not yet been confirmed. Some of them may represent false positives; others may have just been ignored by replication studies due to lack of biological appeal with regard to further analysis.

The filtered list of 62 LRRK2 interactors (i.e., interactors that have been confirmed with at least 2 annotations) was used for enrichment analysis to identify specific biological processes (indicated by GO terms) that LRRK2 is likely to be associated with. These data can be used to infer the function of the LRRK2 interactome. Moreover, based on the assumption that any other protein annotated to these biological processes may interact (directly or indirectly) with LRRK2, these data guide the deduction of additional potential LRRK2 interacting proteins. Enrichment is performed by algorithms developed to recognize if a descriptor (e.g., a GO term associated with biological process or cellular component) is particularly enriched in the list of LRRK2 interactors in comparison with what is expected due to the recurrence of that term in a standard reference list of proteins or genes. This analysis has been performed using Panther (through the GO Consortium portal) and WebGestalt, two freely available online platforms. Since WebGestalt and Panther apply different statistical approaches and use different versions of the GO annotation and ontology datasets they were used to provide 2 independent analyses. The results using both tools were, however, very similar, pointing towards a comparable functional interpretation of LRRK2 interactome; specifically, all the top 10 WebGestalt hits were also present in the Panther enriched list. It has to be noted that a number of 150 genes/proteins is generally suggested as minimum to obtain a significant enrichment (http://geneontology.org/faq/what-minimum-information-include-functional-analysis-paper). Our analysis included just 62 LRRK2 interactors, nevertheless we found statistically significant enrichment with both of the two algorithms we used. This may suggest that the group of proteins we used as input (the LRRK2 interactome) had a low level of background noise which therefore didn’t mask enrichment, and this was probably achieved by using a screened interactome composed only by interactors that have been confirmed in at least one replication study.

In the WebGestalt analysis, 50% of the LRRK2 interactors contributed toward the enrichment of functional terms related with transport/localization, 68% with cell organization and 29% with regulation of kinase activity. Forty percent of the LRRK2 interactors contributed toward the enrichment of terms associated with the cytoskeleton, 37% with cell projections and 31% with vesicles. The enrichment of terms related to the regulation of kinase activity was expected, as LRRK2 is an active kinase and likely to regulate other kinases in cascade. However the LRRK2 interactors contributing to this specific enrichment were also present in the GO term groups of vesicles (50%) and transport/localization (72%) thus suggesting LRRK2’s interactome may regulate kinase functions in vesicle transport. Sixty percent of the LRRK2 interactors involved in the GO term group of cytoskeleton also contributed toward the enrichment of transport. Similarly, 79% of the interactors involved in the GO term group of vesicles were also annotated as involved in transport, thus suggesting that the LRRK2 interactome influences transport processes related with cytoskeleton and vesicles dynamics. These results are in accordance with recent functional literature that sees a role of LRRK2 in vesicle mediated processes like endocytosis (Alegre-Abarrategui et al., 2009), synaptic function (Cirnaru et al., 2014; Parisiadou et al., 2014) and autophagy (Schapansky et al., 2014; Manzoni et al., 2013; Gómez-Suaga et al., 2012). It is of note that the papers referenced in this discussion section are not annotated in the BioGRID neither in the IntAct databases since they are either too recent, or they report only a functional evidence for LRRK2 activity and do not contain PPI records. For this reason they have not contributed toward the enrichment, so they independently support the results.

The Panther analysis identified more descriptive GO terms as enriched, compared with the WebGestalt analysis (Table S4). For this reason, we then analysed details of each single, enriched GO term group. For the most significantly enriched group in Panther, i.e., transport/localization, the 3 most specific terms were “cytoskeleton-dependent intracellular transport,” “cellular protein localization” and “vesicle-mediated transport” thus reiterating the importance of LRRK2 interactome for the process of transport and vesicular trafficking. GTPases and regulators of GTPase activity are a protein group represented in the LRRK2 interactome, it is therefore unsurprising that the GO term GTP catabolic process was enriched in the analysis. Similarly, the enrichment of many terms related to nucleotide catabolism and metabolism was expected due to the presence of ATP/GTPases in the LRRK2 interactome. The function of many kinases is to regulate other kinases therefore the enrichment of terms associated with kinase regulation was expected. It was however impossible to predict from this analysis whether the 11 LRRK2 associated kinases (Table S5) are being regulated by LRRK2 or whether the LRRK2 interactome is regulating LRRK2’s activity. The child terms in the GO term group of cell death specifically indicated the regulation of this process and the special cell death defined as apoptosis. Very interestingly, 3 out of the 5 terms enriched in the GO domain of regulation of mitochondrion organization were related to the regulation of mitochondrial processes in apoptosis (Table S4). The combination of this information may suggest that the key enriched process is apoptosis rather than the general process of cell death, and that it is the apoptotic processes taking place at the mitochondria, rather than mitochondria generic processes, which are relevant to the function of the LRRK2 interactome. When the signalling processes were looked at in detail, all the enriched terms appeared to be very general, with the exception of “Fc receptor signalling pathway” (p-value = 1.32e−11, 21% of LRRK2 interactors contributing to the enrichment of this term). This is a very specific GO term that in the ancestor GO chart is directly connected with cell surface signaling and immune response, and supports the recent findings that microglial Fc receptors have been associated with alpha-synuclein-induced pro-inflammatory signaling in PD (Cao, Standaert & Harms, 2012). There were 8 terms enriched within the immune response domain; 4 of them were parents of the specific term “immune response-regulating cell surface receptor signaling pathway” and thus also have a parental relationship with the enriched GO term “Fc receptor signaling pathway.” It is known that LRRK2 is expressed in the immune system (Thévenet et al., 2011) and that its total level and phosphorylation state are modulated by immune cell stimulation (Moehle et al., 2012; Kubo et al., 2010). Finally, the developmental process domain presented with 15 enriched terms; 5 of them were general terms; the remaining 10 terms were associated with development of neurons and the nervous system. Very little is known about LRRK2 and development; however it has been shown that total LRRK2 levels vary according to different developmental stages (Giesert et al., 2013; Zechel et al., 2010).

In the light of these observations we propose the function of LRRK2 interactome to be associated with transport and trafficking, possibly regulating enzymatic events associated with cytoskeleton and vesicles. However, we recognize that LRRK2 probably has a role as a hub protein, thus controlling many different functions at a time, and this is likely to be one of the reasons why it has proved so challenging to reach a consensus on LRRK2 physiological function. Based on our results we suggest the role of LRRK2 can also be modulated depending on the cell type analysed (i.e., immune versus nervous system) and possibly the developmental stage, so the outcome of wet lab research may depend on the cellular system used to infer interaction. Not only does this give a possible explanation for the number of different functions that are described for LRRK2 in different experimental models; it also suggests a possible reason for the plethora of different diseases LRRK2 is associated with and opens to discussions of a LRRK2-oriented therapy. If LRRK2 regulates different biological processes in different tissues, a hypothetical targeting of LRRK2 during disease may generate unexpected, and unwanted, side effects. It would be beneficial to analyse LRRK2 interactome according on the cell type in which the interaction was reported. Unfortunately at the moment the majority of the annotations come from in vitro studies and cancer cell lines; we do not have enough annotations from primary cultures and physiological model systems to perform cell type specific analysis as yet.

The analysis of the 200 Kbp around the associated SNPs for PD and IBD revealed that some of the proteins encoded in these regions are included in the LRRK2 interactome. The influence of associated SNPs on the control of the regions around them is unknown. They are markers for loci involved in the risk for disease, they may modify the gene in which they actually reside or alternatively regulate multiple genes around them. The presence of 21 LRRK2 interactors around associated SNPs for PD and IBD reinforces the role of LRRK2 in these diseases and provides a system to prioritise candidates to be evaluated in the context of the molecular mechanism of sporadic PD and IBD.

Conclusions

The analysis of the LRRK2 interactome we described above provides an overview of the published research on this protein, and a system in which to identify the interacting proteins with the most reliable supporting experimental evidence. The enrichment analysis of this interactome provides an indication of the pathways and processes LRRK2 can influence within its cellular environment. This analysis indicates that LRRK2 may be involved in more than one function, perhaps depending on the cellular type and developmental stage. Intriguingly, some of the proteins in the LRRK2 interactome have also been suggested to be genetic risk factors for PD and IBD. Our approach can therefore guide future research priorities as it suggests focusing on these interactors and their relationship to LRRK2 and disease. Moreover the multiple interactions and cellular functions associated with LRRK2 suggest caution in using it as a drug target given the potential for off-target problems. Importantly, in silico analyses such as those described above can provide a starting point for further hypothesis driven wet laboratory based investigations, opening up new avenues that may reveal important insights into the biology of LRRK2.

Supplemental Information

Table S1 Merging of 6 problematic records

It contains details regarding the merging of 6 problematic records duplicated differently between the BioGRID and IntAct repositories. Annotations in white has been kept in the final record moved into the merged dataset, annotations in red has been removed. (A).Rationale: (i) Protein Kinase Assay = Biochemical Activity (one of the redundant entries has been removed); (ii) Affinity Capture-Western (BioGRID) is split in Pull Down and Anti-tag Coimmunoprecipitation in IntAct, following the rule of keeping the record with maximum number of annotations, the record in IntAct has been preferred over the one in BioGRID. (B). Rationale: (i) Protein Kinase Assay = Biochemical Activity (one of the redundant entries has been removed). (C). Rationale: (i) Affinity Capture-Western = Affinity Chromatography Technology (one of the redundant entries has been removed). (D).Rationale : (i) Affinity Capture-Western = Anti-tag Coimmunoprecipitation (one of the redundant entries has been removed). (E). Rationale: (i) Affinity Capture-Western/MS = Pull Down (one of the redundant entries has been removed). (F). Rationale: Merged.

Click here for additional data file.

Table S2 LRRK2 interactors annotated 1 time only

It contains details regarding the 207 LRRK2 interactors that were annotated 1 time only.

Click here for additional data file.

Table S3 WebGestalt

It contains details regarding grouping of the terms enriched in WebGestalt.

Click here for additional data file.

Table S4 GO enrichment supported by Panther

It contains details regarding grouping of the terms enriched in Panther. BF, background frequency; SF, sample frequency.

Click here for additional data file.

Table S5 Kinases in the LRRK2 interactome

It contains specifications regarding the kinases in the LRRK2 interactome. Details are from the UniProt database (downloaded on the 9th January 2015).

Click here for additional data file.

Data S1 WebGestalt GO terms enrichment

Results as obtained from the WebGestalt, GO terms enrichment analysis.

Click here for additional data file.

Additional Information and Declarations

Competing Interests

Author Contributions

The authors declare there are no competing interests.

Claudia Manzoni conceived and designed the experiments, performed the experiments, analyzed the data, wrote the paper, prepared figures and/or tables.

Paul Denny wrote the paper, reviewed drafts of the paper.

Ruth C. Lovering analyzed the data, wrote the paper, reviewed drafts of the paper.

Patrick A. Lewis conceived and designed the experiments, contributed reagents/materials/analysis tools, wrote the paper, reviewed drafts of the paper.

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
