# Peer review of "Computational analysis of the LRRK2 interactome"

_PeerJ, doi:10.7717/peerj.778_

## Round 0.1 · original submission · Minor Revisions

Thanks a lot for considering PeerJ to publish your best work. As you will see in the following paragraph, your manuscript was carefully reviewed by two refereed and both of them consider that minor revisions are needed before publication of your manuscript.

Reviewer 1 ·

Basic reporting

Overall the article is well written and easy to understand and follow but there are some minor changes I find necessary. In order of appearance for each section:

Introduction: PD and IBD acronyms should be defined again in the main body of the article, not solely in the abstract, and in general I'd avoid acronyms in the abstract section.

Line 214: …. to present with their contributions, I think this sentence either is missing a word (to be present) or has one too many (to present their contributions).
Line 255: …. is annotated in databases. In which databases? Better if you specify. The sentence starting at the end of this line that goes on for the next three is too long.
Line 281: there is the tendency for proteins to be more frequently examined and therefore annotated, the more they are identified as probable interactors. Rephrase as it is difficult to understand the way it is. There is a tendency for some proteins to be …..., making them more likely to be identified as probable interactors.
Line 284: Sentence starting with “For the many interactors....” is too long, split in several shorter sentences. Explain better why they are not Fps.
Line 290: sentence starting with “These data can be used....” is too long, split into several shorter ones.
Line 299: what do you refer to with “GO annotation and ontology datasets”? Could it be GO versions

Title for figure 1: merge → merged

Very likely I missed some very long sentences that would be made easier to understand and read by splitting them into several shorter sentences.

Experimental design

With regards to the protocol followed in the article I find some aspects that could be improved. Solely two PPIs were included in the study rendering 62 filtered LRRK2 interactors, a number that according to what the authors mention may not be enough to perform the enrichment analysis. This number of interactors could be raised by adding data available in more PPIs repositories such as MINT (http://mint.bio.uniroma2.it/mint/Welcome.do) and STRING (http://string-db.org/) among others.

Another issue is the application of the second filter (filter #2). The authors do not specify from which publication is kept and which is removed, ie the publication with more reported interactors, with less, publications also present in the other database? Proper clarification about this issue will make innecessary the discussion about shared publications in the final dataset used in the enrichment analysis.

Nonetheless, even after these issues, the protocol followed in the article seems appropriate and valid.

Validity of the findings

Even after the issues raised in the comments about the experimental design, the findings explained in the article are validated by additional literature not used in any way in the creation of the dataset employed in the study. Maybe the inclusion of more interactors found in more PPI repositories could provide more robustness to the reported findings, provide more functions after the enrichment study, or provide more specif functions, but until additional interactors are included there is nothing that one can say.

Additional comments

The manuscript describes a computational study of the LRRK3 protein interactome that aims to characterize the function of LRRK2. To perform this study the authors carried out enrichment analysis of the functional annotations described for the known interactors of LRRK2 as found in two well known and established PPI databases. While overall the article meets the requirements for publication there are some aspects whose improvement would greatly benefit the article.

·

Basic reporting

No comments.

Experimental design

No comments.

Validity of the findings

No comments.

Additional comments

This is a very nice study, performing a careful computational analysis of the LRRK2 interactome by collecting data from both BioGRID and IntAct data repositories, combined with GO terms enrichment. The presented bioinformatics approach allows for the development of a vision as to how normal and pathogenic LRRK2 may function in a cellular context, and aids in the design of future hypothesis-driven experimental approaches.
General comment: I wonder whether performing an analysis according to the cell type where a given interaction was reported may shed additional insight into possible cell type-specific interactions? Their analysis confirms that their is a wealth of information regarding LRRK2 self-interaction, and heterologous interactions which clearly implicate LRRK2 in vesicular/membrane trafficking and transport processes. The study, in a non-biased manner, strengthens the evidence for a role of LRRK2 in those cellular events.
Minor comments:
Supplementary Table 2: it would help to put the proteins in alphabetical order.

Line 344: they mention 11 LRRK2-associated kinases. Maybe an additional supplemental table listing those kinases could be useful.

---

## Round 0.2 · accepted · Accept

I'd like to thank to the authors for considering PeerJ as the platform for publication of their best work. This paper makes a compelling effort to perform a computational analysis of an interactome that certainly will attract the attention of PeerJ readers.